Postural control in girls with adolescent idiopathic scoliosis while wearing a Chêneau brace or performing active self-correction: a pilot study

Piątek Elżbieta elzbieta.piatek90@gmail.com 1
Kuczyński Michał 2
Ostrowska Bożena 1
1 Faculty of Physiotherapy, University School of Physical Education in Wroclaw , Wrocław , Poland
2 Faculty of Physical Education and Physiotherapy, Opole University of Technology , Opole , Poland
Daumer Martin
Electronic publication date: 2019 Aug 29
Publication date: 2019
Volume: 7
Electronic Location ID: e7513
Received 2019 Mar 27; Accepted 2019 Jul 18
Copyright: ©2019 Piątek et al.
Copyright year: 2019
Copyright holder: Piątek et al.
License: This is an open access article distributed under the terms of the Creative Commons Attribution License, which permits unrestricted use, distribution, reproduction and adaptation in any medium and for any purpose provided that it is properly attributed. For attribution, the original author(s), title, publication source (PeerJ) and either DOI or URL of the article must be cited.
License URL: https://creativecommons.org/licenses/by/4.0/

Keywords: Adolescent idiopathic scoliosis, Chêneau brace, Active self-correction, Postural control

Funding: The authors received no funding for this work.

==============================
Background

It is known that adolescent idiopathic scoliosis (AIS) is often accompanied by balance deficits. This reciprocal relationship must be taken into account when prescribing new therapeutic modalities because these may differently affect postural control, interacting with therapy and influencing its results.

Objective

The purpose was to compare postural control in girls with AIS while wearing the Chêneau brace (BRA) or performing active self-correction (ASC) with their postural control in a quiet comfortable stance.

Methods

Nine subjects were evaluated on a force plate in three series of two 20-s quiet standing trials with eyes open or closed; three blocks were randomly arranged: normal quiet stance (QST), quiet stance with BRA, and quiet stance with ASC. On the basis of centre-of-pressure (COP) recordings, the spatial and temporal COP parameters were computed.

Results and Discussion

Performing ASC was associated with a significant backward excursion of the COP mean position with eyes open and closed (ES = 0.56 and 0.65, respectively; p < 0.05). This excursion was accompanied by an increase in the COP fractal dimension (ES = 1.05 and 0.98; p < 0.05) and frequency (ES = 0.78; p = 0.10 and ES = 1.14; p < 0.05) in the mediolateral (ML) plane. Finally, both therapeutic modalities decreased COP sample entropy with eyes closed in the anteroposterior (AP) plane. Wearing BRA resulted in ES = 1.45 (p < 0.05) while performing ASC in ES = 0.76 (p = 0.13).

Conclusion

The observed changes in the fractal dimension (complexity) and frequency caused by ASC account for better adaptability of patients to environmental demands and for their adequate resources of available postural strategies in the ML plane. These changes in sway structure were accompanied by a significant (around 25 mm) backward excursion of the mean COP position. However, this improvement was achieved at the cost of lower automaticity, i.e. higher attentional involvement in postural control in the AP plane. Wearing BRA may have an undesirable effect on some aspects of body balance.

Introduction

Adolescent idiopathic scoliosis (AIS) is one of the common spinal deformities observed during adolescence, affecting 2–4% of individuals aged 10–16 years (Hawasli, Hullar & Dorward, 2015). AIS can introduce wide-ranging dysfunction in important bodily systems and organs (Simoneau et al., 2006; Wang et al., 2011; Weinstein et al., 2008). Research in the recent decade has attempted to determine if AIS patients show proprioceptive and somatosensory impairment. Several studies on AIS prognosis have reported abnormal somatosensory function, balance control, and proprioception (Guo et al., 2006). Numerous researchers present that standing balance assessments show greater postural instability in AIS patients compared with age-matched controls (Gaudreault et al., 2005; Karimi, Kavyani & Kamali, 2016; Chow et al., 2006). The cited literature suggests that young patients with scoliosis are more susceptible than healthy individuals to external perturbations and have greater difficulty in righting the body after disturbed balance (Karimi, Kavyani & Kamali, 2016).

The underlying cause of impaired postural control is considered multifactorial albeit strongly associated with vestibular dysfunction and defects in certain structures of the central nervous system (Hawasli, Hullar & Dorward, 2015; Gauchard et al., 2001; Byl et al., 1997). As the medial vestibulospinal tract controls the axial muscles (Pialasse et al., 2013), changes in the brain stem or sensorimotor cortex during the critical preadolescent and adolescent period of growth can impair sensorimotor integration and therefore lead to inappropriate trunk muscle activities, spine deformation, and greater instability. Hence, it is hypothesized that reduced balance control in AIS patients may be caused by either impaired vestibular information transfer or sensorimotor processing (Shi et al., 2013).

Two of the most common non-operative treatments prescribed for AIS are physiotherapy via active self-correction (ASC) and orthotic braces (Coillard et al., 2003; d’Amato, Griggs & McCoy, 2001; Katz & Durrani, 2001). ASC is a series of movements of realignment as a whole that the patients autonomously perform in order to reduce the scoliotic curves. This exercise has to be implemented as much as possible in three dimensions, which sometimes makes the ASC movement not easy to be understood and also not easy to be completed by the patient (Pizzetti et al., 2010). Bracing is generally recommended for skeletally immature patients with Cobb angles of 25–45° so as to halt further progressive curvature and provide passive correction (Chow, Leung & Holmes, 2007). Within the group of rigid Thoraco-Lumbo-Sacral Orthoses (TLSO), the Chêneau brace (BRA) is most widely used in Poland (Zaborowska-Sapeta et al., 2011). However, the clinical effects of bracing on postural balance in the AIS-afflicted population have only been addressed by a few studies, which report inconsistent findings. In the available literature, there is evidence that a bracing intervention can improve postural stability and balance in the sitting position (Smith & Emans, 1992). One investigation confirmed that bracing did not impede stability or balance during upright stance on a solid surface but significantly increased the centre-of-pressure (COP) sway area, displacement, and mediolateral amplitude in patients standing on an unstable surface (Chow, Leung & Holmes, 2007). Other studies have reported no significant clinical effects, among others, no changes in COP displacement or sway area, after a 4-month application of the Boston bracing system compared with controls (Sadeghi et al., 2008). Instead, greater postural stiffness in the anteroposterior direction and reduced mediolateral balance control were observed. Another study demonstrated improved postural balance after 4 months of bracing in young AIS patients (Paolucci et al., 2013; Khanali et al., 2015).

A significant issue is that the vast majority of studies have focused solely on the effects of BRA-induced alignment on postural control whereas few have addressed autocorrective-based approaches (three-dimensional ASC). The diagnostic and therapeutic recommendations published by the Society on Scoliosis Orthopaedic and Rehabilitation Treatment (SOSORT) show consensus on the use of autocorrection exercises and the efficacy of this intervention (Negrini et al., 2018). To the authors’ knowledge, there are no studies that compare postural control in girls with AIS while wearing BRA or performing ASC. Therefore, the aim of this study was to compare postural control in girls with AIS while wearing BRA or performing ASC with their postural control in a quiet comfortable stance. We hypothesized better efficacy of ASC in supporting postural control.

Materials & Methods

Participants

Nine post-menarche female AIS patients aged 11–16 years (age: 14  ± 1.48 years, weight: 47.71  ± 5.61 kg, height: 161.5  ± 8.81 cm, Cobb angle: 35.6  ± 8.9°) from a local therapeutic rehabilitation centre participated in the study. All subjects had normal vision. Individual patient characteristics including scoliotic curvature details are presented in Table 1. The inclusion criteria for the participants were diagnosis of AIS by an independent physician and receiving conservative treatment: BRA and physiotherapeutic scoliosis-specific exercises (PSSE) recommended by SOSORT (Negrini et al., 2018). Patients who had history of spine surgery, musculoskeletal, or neurological disease, or any spinal pathology not comorbid with AIS were excluded from the study. All participants received brace treatment for a minimum of 2 months (3.56 ± 1.42 months) with a dosage of 20 hours/day; the brace was removed for personal hygiene, exercise, or delineated rest periods. Each BRA was custom designed for three-dimensional curve correction by an experienced orthotics specialist. Figures 1 and 2 illustrate an exemplary BRA used in the study. All patients knew their own ASC movements which reduced scoliotic curves. These included different types of movements: (1) controlled self-elongation, having regard to the sagittal plane; (2) correction of the primary curve in the frontal plane; (3) correction of contiguous curves in the frontal plane; (4) correction of the primary curve in the horizontal plane.

Table 1 Anthropometric and scoliotic curvature characteristics of the participants.

No.	Sex	Age [years]	Body mass [kg]	Body height [cm]	Risser sign	AIS curve type	Cobb angle [°]	Apex	
1	Female	14	45.6	163.5	3	Right thoracic/left lumbar	30/32	T8/L2	
2	Female	14	53.5	169	3.5	Right thoracic/left lumbar	28/40	T8/L2	
3	Female	12.5	52.2	163.5	3.5	Right thoracic/left lumbar	16/25	T7/L1	
4	Female	15	51.8	170.5	2	Right thoracic/left lumbar	18/35	T9/L3	
5	Female	15	53.1	171.5	3	Right thoracic/left lumbar	37/20	T8/L1	
6	Female	16	48.8	158	4	Right thoracic/left lumbar	32/30	T8/L1	
7	Female	14.5	40.8	153	1.5	Right thoracic/left lumbar	55/20	T7/L1	
8	Female	11	38	144.5	1	Right thoracic/left lumbar	26/20	T8/L1	
9	Female	14	45.6	160	3	Right thoracic/left lumbar	38/18	T9/L3	

Figure 1 The Chêneau brace used in this study (back).

Figure 2 The Chêneau brace used in this study (front).

Written informed consent was obtained from all participants and their parent(s) or legal guardian(s). The study goals, procedures, and methods were explained in full, and the subjects were informed that they could withdraw at any time. The study was approved by the Senate Research Ethics Committee at the University School of Physical Education in Wroclaw, Poland (the approval number: 35/2016).

Methods

All procedures were performed in laboratory conditions. Postural control was assessed barefoot on a Kistler force platform (Kistler 9281CA, Winterthur, Switzerland). Two-dimensional horizontal coordinates of the COP data were recorded for 20 s at a sampling frequency of 100 Hz. Three bipedal quiet standing trials were performed: (1) QST—normal quiet stance: standing upright with a neutral and comfortable stance with arms relaxed at the sides; (2) ASC: standing upright with autocorrection; on the ‘correction’ command, the participant performed ASC; and (3) BRA: standing upright wearing BRA.

Each trial was performed in eyes-open and eyes-closed conditions. The foot position (5 cm apart) was standardized on the surface to ensure repeatability across trials and participants. Task order was counterbalanced with 1 min of rest provided after each trial in order to minimize the effects of fatigue or hysteresis (Chow, Leung & Holmes, 2007). The participants were instructed to stand as motionless as possible. Data acquisition began when the subject signalled they were ready. Relevant COP outcome measures were determined separately for the mediolateral (ML) and the anteroposterior (AP) direction. These were:

• COP SD [mm]—standard deviation of COP displacement from mean COP;

• COP sway range [mm]—difference between the maximum and minimum value of the COP;

• COP mean [mm]—mean value of the COP position;

• COP mean velocity [mm/s]—COP excursion divided by trial time;

• COP fractal dimension—a non-linear dynamic parameter of COP where the greater the fractal dimension, the better the postural system adapts to changes;

• COP sample entropy—a non-linear dynamic parameter of COP where greater entropy (higher COP irregularity) suggests less attentional resources devoted to balance maintenance (greater automaticity);

• COP frequency [Hz]—frequency of COP, indicative of the involvement of the neural system in postural regulation.

Statistical analysis

Data were processed with the Statistica 12.0 software package (StatSoft, Tulsa, OK, USA). The data met the criteria of normal distribution for all parameters of COP measures. Thus, to evaluate the possible effects and interactions of vision (eyes open, eyes closed) and posture (quiet standing, ASC, BRA), a 2 × 3 repeated analysis of variance (ANOVA) was conducted for all parameters of the COP in the ML and the AP planes separately. To assess differences between the three groups, the post-hoc Fisher’s least significant difference (LSD) test was used. The level of significance was set at p < 0.05. Effect size (ES) was calculated to determine the effects of ASC and BRA on postural control compared with quiet standing. An effect size of 0.3 is a small effect, 0.5 is a moderate effect, and 0.8 is a strong effect.

Results

In the AP plane, the COP mean position showed the main effect of posture (F[2, 16] = 3.73; p = 0.0047). The LSD test pointed at a significant backward shift of the COP mean position in both eyes open (p = 0.004) and eyes closed (p = 0.0002) (Fig. 3). The traditional measures of the COP dispersion displayed the main effects of vision in the AP plane for COP variability (F[1, 8] = 17.78; p = 0.003), range (F[1, 8] = 18.33; p = 0.0003), and velocity (F[1, 8] = 9.57; p = 0.001). Eyes closure increased the values of these three latter parameters. In addition, there was a vision × posture interaction (F[2, 16] = 3.89; p = 0.042) for range only, which arose because brace differently affected this parameter: the range increased during eyes closed (p = 0.003) and remained unchanged during eyes open. The COP sample entropy in the AP plane displayed the main effect of vision (F[1, 8] = 6.87; p = 0.031), indicating lower values with eyes closed. Post-hoc analysis revealed that using a brace resulted in a significantly lower (p = 0.02) sample entropy during eyes closed (Fig. 4). Interestingly, there were no differences between ASC and quiet standing.

Figure 3 Mean (SD) values of COP mean position in the AP plane in the backward directions.

QST, quiet standing; ASC, active self-correction; BRA, Chêneau brace. White column, eyes open; shaded column, eyes closed. Asterisks indicate a significant (p < 0.05) post hoc difference with respect to QST.

Figure 4 Mean (SD) values of Samle entropy in the AP plane.

QST, quite standing; ASC, active self-correction; BRA, Chêneau brace. White column, eyes open; shaded column, eyes closed. Asterisks indicate a significant (p < 0.05) post hoc difference with respect to QST.

In the ML plane, the results of ANOVA proved the main effect of posture on the COP fractal dimension (F[2, 16] = 6.05; p = 0.011) and sway frequency (F[2, 16] = 10.7; p = 0.001). The post-hoc LSD test indicated the highest values of the COP fractal dimension during ASC with eyes open (p = 0.01) and eyes closed (p = 0.03) compared with the brace and quiet standing (Fig. 5). The post-hoc analysis of frequency did not reveal significant changes (Fig. 6).

Figure 5 Mean (SD) values of COP fractal dimension in the ML plane.

QST, quiet standing; ASC, active self-correction; BRA, Chêneau brace. White column, eyes open; shaded column, eyes closed. Asterisks indicate a significant (p < 0.05) post hoc difference with respect to QST.

Figure 6 Mean (SD) values of COP frequency in the ML plane.

QST, quite standing; ASC, active self-correction; BRA, Chêneau brace. White column, eyes open; shaded column, eyes closed.

The means (SD) of all dependent variables including effect size values over 0.5 are shown in Table 2.

Discussion

The purpose of this study was to compare postural control in girls with AIS while wearing BRA or performing ASC with their postural control in a quiet comfortable stance. In view of the fundamental difference between the two therapeutic approaches, i.e., BRA and ASC, we hypothesized better efficacy of ASC in supporting postural control. This prediction was mainly based on the expected releasing degrees of freedom and facilitating exploratory function by ASC as opposed to biomechanical constraints imposed by the brace, rendering the hip and trunk joints practically inflexible. On the other hand, releasing degrees of freedom has been suggested as an important characteristic of motor learning, which is necessary when exploring solutions for a novel task (Sternad, 2018).

Table 2 Mean ± SD in AP and ML COP outcome measures for posture trials.

Direction	Variable	Eyes open	Eyes closed	
		Quiet standing	Active self-correction	Chêneau brace	Quiet standing	Active self-correction	Chêneau brace	
Anteroposterior	COP SD [mm]	3.89 ± 1.30	4.09 ± 1.87	3.92 ± 1.36	3.97 ± 1.46	5.93 ± 3.06*,b	6.25 ± 2.8*1b	
COP sway range [mm]	18.96 ± 5.61	19.07 ± 9.12	17.50 ± 5.54	19.73 ± 8.86	26.88 ± 12.80	30.23 ± 12.85	
COP mean velocity [mm/s]	12.12 ± 2.11	13.14 ± 3.71	11.58 ± 2.26	14.60 ± 3.83	17.32 ± 6.43	15.86 ± 4.23	
COP frequency [Hz]	0.30 ± 0.11	0.32 ± 0.07	0.29 ± 0.18	0.40 ± 0.13	0.39 ± 0.12	0.37 ± 0.19	
COP fractal dimension	1.43 ± 0.06	1.43 ± 0.03	1.41 ± 0.07	1.45 ± 0.05	1.44 ± 0.05	1.43 ± 0.05	
COP sample entropy	1.23 ± 0.26	1.27 ± 0.28	1.18 ± 0.27	1.30 ± 0.26	1.10 ± 0.29*,a	0.95 ± 0.22*,b	
COP mean [mm]	24.89 ± 28.81	43.1 ± 35.2a	23.1 ± 26.1	16.28 ± 25.87	43.86 ± 51.24*,a	16.68 ± 23.77	
Mediolateral	COP SD [mm]	3.33 ± 1.33	3.11 ± 1.40	3.26 ± 1.32	3.81 ± 1.96	3.96 ± 1.84	3.65 ± 1.48	
COP sway range [mm]	16.52 ± 6.35	16.65 ± 7.54	15.88 ± 4.89	18.51 ± 9.69	19.74 ± 8.46	18.31 ± 8.01	
COP mean velocity [mm/s]	9.35 ± 2.76	11.13 ± 4.42	9.25 ± 2.11	11.14 ± 5.47	13.91 ± 5.27	11.15 ± 4.02	
COP frequency [Hz]	0.33 ± 0.11	0.42 ± 0.12*,a	0.39 ± 0.14	0.35 ± 0.11	0.47 ± 0.10*,b	0.39 ± 0.08	
COP fractal dimension	1.43 ± 0.06	1.48 ± 0.03*,b	1.40 ± 0.06	1.44 ± 0.06	1.49 ± 0.04*,b	1.45 ± 0.02	
COP sample entropy	1.06 ± 0.34	1.22 ± 0.22	1.07 ± 0.31	1.03 ± 0.30	1.09 ± 0.22	1.05 ± 0.18	
COP mean [mm]	−4.26 ± 13.28	−3.9 ± 16.0	−1.26 ± 8.37	−4.32 ± 10.94	−0.15 ± 9.32	−2.03 ± 11.00	
Notes.

* Significant difference at p < 0.05.

a ES > 0.5.

b ES > 0.8.

To find the effects of the investigated postures on the possible preferential use of different sensory afferents, we examined the subjects with the manipulation of visual input. Such an experimental protocol provides unique opportunities to reveal how sensory and motor signals are integrated to control the upright body (Paillard, Bizid & Dupui, 2007; Kabbaligere, Lee & Layne, 2017; Rasman et al., 2018). However, the results failed to find any interactions between the postural condition and vision in any of the sway indices. This means that changes in postural control between ASC and BRA observed in this study cannot be explained by differential effects of the investigated postures on the quality of sensory inputs. Instead, they must have been caused by some enhancement in sensory integration and internal representation of verticality. The latter proposition is consistent with some other studies on AIS (Gauchard et al., 2001; Simoneau et al., 2006; Catanzariti et al., 2014), as well as on stroke or vestibular pathology (Pérennou et al., 2008; Borel et al., 2008).

In contrast to the weak interactions, there were strong main effects of vision on several postural indices. Eyes closure deteriorated postural performance, as evidenced by the increased COP variability and mean velocity. The same manipulation decreased COP entropy, indicating the need for more conscious and less automatic postural control with occluded vision. Additionally, eyes closure resulted in an increased COP frequency, suggesting that a higher rate of the exploratory sway function is required when some of the normally available sensory inputs are suppressed. The results of these manipulations add to our understanding of the differing effects of BRA and ASC on the COP parameters in patients with AIS.

With regard to the specific results of this study, aimed at comparing the role of BRA and ASC in postural control, four findings seem of particular value. First, ASC resulted in a significant backward shift of the COP mean position on a hard surface in both visual conditions. This is equivalent to a very similar mean backward shift of the body centre of mass. Second, this shift was accompanied by an increase in the ML fractal dimension and frequency. Third, both therapeutic modalities increased sway amplitude only in the AP plane with eyes closed. Furthermore, in the AP plane with eyes closed, sway entropy decreased with BRA and to a lesser extent with ASC.

Mean COP position is seldom computed and analysed except in subjects standing at heights (Carpenter et al., 2001) and in older persons (Jbabdi, Boissy & Hamel, 2008), where fear of falling significantly contributes to increasing the distance between the limits of stability and the COP mean position. The fact of ignoring this index in the majority of studies probably indicates a high consistency in adopting this position regardless of various sensory manipulations and accounts for a presence of a strong set-point used by the central nervous system in controlling erect stance. Therefore, in our subjects, this backward shift of the mean COP position that resulted from the ASC posture must be interpreted as a relevant adjustment in postural control. By virtue of the unexpected emergence and its impressive magnitude only, the backward displacement of the COP cannot be instantly classified as a beneficial modification of posture. Importantly, however, this modification was accompanied by an increase in the COP fractal dimension and frequency, which manifest advantageous changes in the postural control system. The fractal dimension is regarded as a measure of complexity of the investigated time-series and, within some limits, its higher values account for improvements in the overall postural strategies and in the better adaptability to novel postural challenges, in particular. A higher COP complexity reflects the ability of the central nervous system to use a variety of postural strategies to maintain stable stance (Cone, Goble & Rhea, 2017).

As documented by Cone, Goble & Rhea (2017), balance training that focused at a better sensory reweighting resulted in an increased complexity of sway. In the same vein, Casabona et al. (2016) showed a larger fractal dimension in ballet dancers than in non-dancers and concluded that this difference might indicate a rearrangement of sensory integration and motor adaptation necessary to meet the particular demands of selected ballet performances. This is in line with our understanding of the results of this study. Lower values of complexity found in AIS in habitual postures and while wearing braces may indicate the presence of biomechanical (and also psychological) constraints impeding the selection of sensorimotor strategies which might lead to optimal postural behaviour. The situation was quite different after the patients learned to adequately use the ASC posture. The mechanically and habitually imposed constraints were, at least partially, removed, as reflected by larger sway complexity, which indicated an improved activity of the sensorimotor system in the integrating available sensory inputs and of its capacity to perform the exploratory function.

It is important to point out that in contrast to the apparent gain in the organization of the ASC postural control in the ML plane, the respective changes in the AP plane revealed an unexpected outcome. While using ASC with eyes closed, our patients decreased sway entropy, which reflected more attention invested in postural corrections, characteristic of conscious or deliberate involvement into this process. By doing so, they constrained the normal automatic mode of operation and this interaction resulted in decreased postural performance, which was manifested by increases in sway variability. Although these changes in COP entropy are much lesser than those caused by BRA in the same condition, they may account for some problems that the AIS patients experienced in maintaining the backward shift associated with the new body alignment while performing ASC. Although these undesirable reactions occurred only in trials with eyes closed, when limited sensory input made postural control more challenging, their presence may be associated with a suboptimal design of the self-corrected posture or inadequate adherence of subjects to the rules established by therapists. Such questions have seldom, if ever, been asked in relation to postural performance of subjects with AIS. The results of this study suggest that a regular application of posturography in AIS patients may help answer these questions and shed light on how to improve the treatment of this disease.

The main limitation of this pilot study is a small number of participants, which may have concealed significant changes in postural performance brought about by BRA or ASC. Thus, in addition to statistical significance, we also focused on the magnitude of effect sizes when interpreting the study results. Nevertheless, the subsequent conclusions should be regarded with caution unless confirmed by a replication study with an adequate sample size. The results of this study may provide support as the input data for sample size calculation.

The second limitation concerns the lack of postural control assessment at some intermediate points between the ASC practice and bracing onset and the final test. The large backward shift of the mean COP position did not, most likely, occur during this final test but rather resulted from a continuous process of postural adjustments. One can only speculate that the latter modification of the gravity line alignment tends to optimize its position as a set-point for controlling erect stance. Owing to its direct relationship to poor posture in scoliosis, further research on this issue is warranted. Along the same lines, the investigation into the acute effects of new braces or postures on body stability may be helpful in clinical practice, facilitating the trade-off between body alignment and postural control.

Conclusions

This study compared postural control in girls with AIS while wearing BRA or performing ASC with their postural control in a quiet comfortable stance. We predicted better control during self-correction because its studious and diligent character seems to promote the exploratory function of postural sway, necessary in exploiting new strategies, as opposed to constraints imposed by the brace. Our hypothesis was confirmed by the increase in the COP complexity and frequency in the ML plane while performing active self-correcting movements in comparison with habitual standing. Such changes in complexity and frequency are supposed to account for better adaptability of patients to environmental demands and for their adequate resources of available postural strategies. However, this apparent improvement was achieved at the cost of lower automaticity, i.e., higher attentional involvement in postural control in the AP plane. Notably, the observed changes in sway were accompanied by a significant (around 25 mm) backward shift of the mean COP position, which, we believe, may indicate a relevant adjustment in postural control.

Additional Information and Declarations

Competing Interests

Author Contributions

Human Ethics

Data Availability

The authors declare there are no competing interests.

Elżbieta Piątek conceived and designed the experiments, performed the experiments, analyzed the data, contributed reagents/materials/analysis tools, prepared figures and/or tables, authored or reviewed drafts of the paper, approved the final draft.

MichałKuczyński and Bożena Ostrowska conceived and designed the experiments, analyzed the data, contributed reagents/materials/analysis tools, prepared figures and/or tables, authored or reviewed drafts of the paper, approved the final draft.

The following information was supplied relating to ethical approvals (i.e., approving body and any reference numbers):

The study was approved by the Senate Research Ethics Committee at the University School of Physical Education in Wroclaw, Poland (Ethical Application Ref: 35/2016).

The following information was supplied regarding data availability:

Data is available at Zenodo:

Elżbieta Piątek, MichałKuczyński, & Bożena Ostrowska. (2019). The effects of Chêneau bracing and active self-correction on postural balance in patients with adolescent idiopathic scoliosis. http://doi.org/10.5281/zenodo.2604776.

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
