# Peer review of "Postural control in girls with adolescent idiopathic scoliosis while wearing a Chêneau brace or performing active self-correction: a pilot study"

_PeerJ, doi:10.7717/peerj.7513_

## Round 0.1 · original submission · Major Revisions

Please try to deal with the three reviewer's comments, in particular the concerns of reviewer #2 and #3 are critical, and it should be made clear that the scope of the study is a pilot study.

Reviewer 1 ·

Basic reporting

1. The manuscript is well written and is clear. There are, however, a few typos that need to take care of:
a. Line 72: “reallignemnt” should be” realignment”; also “patients autonomously performs” should be “patients autonomously perform”.
b. Line 151, 170, Figure 3, 4, 5 and 6: “quite” should be “quiet”
c. Line 251-2: “the respective changes in the AP plane hinted at the uncalled-for outcome” – do not quite understand. Suggest to rephrase it.
d. Line 265:”shed light”. Would “shed light on” more appropriate?
e. Figure 3: Some words missing after the phrase “post hoc dif….”
f. Figure 4: “samle” should be “sample”
g. Figure 5 Some words missing after the phrase “post hoc difference with respect …”

Experimental design

No Comment

Validity of the findings

No comment

Additional comments

The findings are valid, albeit the small size of the samples. The group is not a homogeneous group, with some having possibly Lenke 1 and others Lenke 3 curves. The authors are suggested to comment if there are differences between patients with the different types of curves and if the outcome measures vary with the size of the curves.

Reviewer 2 ·

Basic reporting

The intuition behind the design of the study is original and is interesting. However, there are some points that need to be clarified.
I would advise you on the manuscript revised by a native speaking, as there are several grammar mistakes in the text, and some sentences are hard to understand.
no other comment

Experimental design

A weakness of the work is certainly the small sample size limited to the female gender only.
1. Furthermore, going to look at the tables (table 1) with reference to the Cobb data, a non-uniformity of the sample would appear to emerge.
2. Moreover, in the exclusion criteria, no mention is made of how to consider Visus defects: did any of the subjects have visual defects with lens correction? This could represent a bias.
3. It is advisable to report the value of the P, especially where significant, in the Report of the Cop values in the three conditions indicated in material and methods in Table 2: "(1) QST- standing upright with a neutral and comfortable stance with the arms relaxed at the sides, (2) ASC- standing upright with autocorrection where on the command “correction” the participant performed an active self-correction, and (3) BRA- standing upright wearing the Chêneau brace";this allows the reader to have a more immediate perception of how the data moves with respect to which subgroup directly from the table.

Validity of the findings

Considering the small sample number the conclusions should be softened.
Furthermore, had the patients been previously instructed on "how to correct the posture"? Because each of us can have different postural correction strategies and especially in scoliosis, self-correction may also depend on the presence, or not, of a bad alignment of the column on the sagittal plane.

Additional comments

Dear Authors
the idea is original but the paper has as its limit a too small sample size: keeping only nine female subjects, the description of the sample have to be more detailed (as previous prescriptions, other physiotherapy, sports activities etc ...), presence of other postural associations with scoliosis if present or not as the plantar foot support, the alignment of the column on the sagittal plane etc ...
There are some points to be clarified and explained better:
-I would advise you on the manuscript revised by a native speaking, as there are several grammar mistakes in the text, and some sentences are hard to understand.
- (table 1) with reference to the Cobb data, a non-uniformity of the sample would appear to emerge.
-in the exclusion criteria, no mention is made of how to consider Visus defects: did any of the subjects have visual defects with lens correction? This could represent a bias.
-report the value of the P, especially where significant, in the Report of the Cop values in the three conditions indicated in material and methods in Table 2
-had the patients been previously instructed on "how to correct the posture"?
- Considering the data, the conclusions should be softened.

Reviewer 3 ·

Basic reporting

• The title of your study is not consistent with objective and needs to reflect the content.

• Bracing has been identified and discussed in the literature (lines 75 -76). However, you did not mention about Cheneau bracing.

• English language needs improving throughout.

Experimental design

• The aim of the study is not reflect the content.

• You recruited 9 post-menarche female with AIS. However, you did not give details about sample size calculation. In addition, you did not mention why you recruited only post-menarche female (line 103). I am not clear why you recruit participants receiving brace treatment 2 months.

• Could you provide the reason why you need recruit participants receiving brace treatment 2 months? (line 109)

• Could you describe how participants know their ASC, please? and how long they practice ASC? And how to perform ASC (lines 113-114).

Validity of the findings

• You have raised the point about the fundamental difference between the two therapeutic approaches: passive and active mode of operation (lines 186- 187). However, you did not provide details in this 2 therapeutic approaches.

• As your study compared the effects of Cheneau brace and Active self correction. However, you have discussed effects of general bracing on postural balance. Could you discuss specific brace (Cheneau brace) on postural balance.

• “First, the ASC resulted in a significant backward excursion of the COP mean position on hard surface in both visual conditions” In this sentence, could you explain more about increase or decrease backward excursion of the COP?

• The limitation of the study ( lines 269 – 271) and further study (286-290) should be in discussion

Annotated reviews are not available for download in order to protect the identity of reviewers who chose to remain anonymous.

---

## Round 0.2 · accepted · Accept

I would like to ask you to implement the suggestions of the reviewer that are in the annotaed manuscript.

Reviewer 3 ·

Basic reporting

no comment

Experimental design

no comment

Validity of the findings

no comment

Annotated reviews are not available for download in order to protect the identity of reviewers who chose to remain anonymous.